# Urinary N-acetylglucosaminidase in People Environmentally Exposed to Cadmium Is Minimally Related to Cadmium-Induced Nephron Destruction

**DOI:** 10.3390/toxics12110775

**Published:** 2024-10-25

**Authors:** Soisungwan Satarug

**Affiliations:** Centre for Kidney Disease Research, Translational Research Institute, Woolloongabba, Brisbane, QLD 4102, Australia; sj.satarug@yahoo.com.au

**Keywords:** albuminuria, cadmium, β_2_-microglobulin, eGFR, mediation analysis, N-acetyl-β-D-glucosaminidase

## Abstract

Exposure to even low levels of the environmental pollutant cadmium (Cd) increases the risk of kidney damage and malfunction. The body burden of Cd at which these outcomes occur is not, however, reliably defined. Here, multiple-regression and mediation analyses were applied to data from 737 non-diabetic Thai nationals, of which 9.1% had an estimated glomerular filtration rate (eGFR) ≤ 60 mL/min/1.73 m^2^ (a low eGFR). The excretion of Cd (E_Cd_), and renal-effect biomarkers, namely β_2_-microglobulin (E_β2M_), albumin (E_alb_), and N-acetylglucosaminidase (E_NAG_), were normalized to creatinine clearance (C_cr_) as E_Cd_/C_cr_ E_β2M_/C_cr_, E_alb_/C_cr,_ and E_NAG_/C_cr_. After adjustment for potential confounders, the risks of having a low eGFR and albuminuria rose twofold per doubling E_Cd_/C_cr_ rates and they both varied directly with the severity of β_2_-microglobulinuria. Doubling E_Cd_/C_cr_ rates also increased the risk of having a severe tubular injury, evident from E_NAG_/C_cr_ increments [POR = 4.80, *p* = 0.015]. E_NAG_/C_cr_ was strongly associated with E_Cd_/C_cr_ in both men (β = 0.447) and women (β = 0.394), while showing a moderate inverse association with eGFR only in women (β = −0.178). A moderate association of E_NAG_/C_cr_ and E_Cd_/C_cr_ was found in the low- (β = 0.287), and the high-Cd body burden groups (β = 0.145), but E_NAG_/C_cr_ was inversely associated with eGFR only in the high-Cd body burden group (β = −0.223). These discrepancies together with mediation analysis suggest that Cd-induced nephron destruction, which reduces GFR and the tubular release of NAG by Cd, involves different mechanisms and kinetics.

## 1. Introduction

Chronic kidney disease (CKD) is a progressive disease with significant morbidity and mortality [1,2]. Globally, death from CKD rose from the 13th highest cause of death in 2000 to the 10th highest cause of death in 2019, and has now reached epidemic proportions, projected to be the fifth leading cause of years of life lost by 2040 [2]. CKD is diagnosed when the estimated glomerular filtration rate (eGFR) falls below 60 mL/min/1.73 m^2^, or there is albuminuria for at least 3 months [3,4,5]. In its early stage, CKD is largely asymptomatic. This makes its early detection difficult and the initiation of early treatment, which can significantly prevent CKD progression, limited [4,5].

Exposure to the environmental pollutant cadmium (Cd) is inevitable for most people because of its ubiquitous presence in the human diet, evident from a food safety monitoring program called the Total Diet Study [6,7,8]. Polluted air and active and passive smoking are additional Cd exposure routes [9,10,11]. Cd has no nutritional value or physiological role, and its health impact has long been underappreciated. Concerningly, a tolerable intake of Cd at 0.83 μg per kg body weight per day (58 µg per day for a 70 kg person), set by the Joint FAO/WHO Expert Committee on Food Additives and Contaminants (JECFA) [12] is not low enough to be without an appreciable health risk. The JECFA “tolerable” intake level of Cd assumed a nephrotoxicity threshold of Cd excretion at 5.24 μg/g of creatinine [12]. However, it is now known that most of the excreted Cd originates from injured or dying tubular cells, and the excretion of Cd reflects injury at the present time, not the risk of injury in the future [13].

Long-term exposure to Cd has been repeatedly linked to an increased risk of CKD in the general population worldwide [6,14]. Studies from Japan reported the absorption rates of Cd among women to be as high as 24–45% [15,16]. Zinc status and body iron storage status are key determinants of the body burden of Cd [17]. Cd accumulates mostly in the kidney tubular epithelium [18,19,20]. Here, it impairs mitochondrial function and promotes the generation of reactive oxygen species (ROS) [21], disrupting calcium homeostasis by the endoplasmic reticulum with resultant tubular cell death [22,23]. Of interest, a recent study has shown that ferroptosis and injury to kidney tubular cells due to Cd was through its induction of heme oxygenase-1 [24].

Ample evidence suggests that exposure to low concentrations of Cd increases the risks of both low eGFR and albuminuria [14,25], but research into the mechanisms by which Cd induces these pathologies is scarce. The present study has two major aims. The first aim is to explore a cause–effect inference of Cd exposure, tubular injury, and a declining eGFR through simple mediation analysis. The second aim is to determine the body burden level of Cd at which these outcomes occur, using data on urinary excretion of β_2_-microglobulin (β_2_M), albumin, and N-acetyl-β-D-glucosaminidase (NAG) as kidney effect biomarkers. Urinary NAG emanates from injured or dying tubular cells, and thus its excretion reflects kidney injury from any cause [26,27].

The protein β_2_M is filtered freely by the glomeruli and is reabsorbed almost completely by the kidney’s tubular cells [28,29]. Cd has been shown to cause a reduction in the tubular reabsorption of β_2_M [30], and increased β_2_M excretion has been used as an indicator of tubulopathy. In comparison, the protein albumin is not normally filtered by glomeruli due to its large molecular weight and its negative charge [31,32,33]. By means of transcytosis through endothelial cells and podocyte foot processes, albumin reaches the proximal tubular lumen [34] and is reabsorbed and returned to circulation by three major mechanisms: fluid-phase endocytosis, megalin/cubillin receptor-mediated endocytosis, and transcytosis [31,32,33].

## 2. Materials and Methods

### 2.1. Study Subjects

The present study was conducted following the principles outlined in the Declaration of Helsinki. To obtain a group of subjects with a wide range of Cd exposure levels appropriate for dose–response analysis, study subjects included residents of Bangkok (*n* = 200) and Mae Sot district, Tak province, where Cd contamination was endemic (*n* = 537). The paddy soil samples from the Mae Sot district had Cd concentrations above the standard (0.15 mg/kg), and samples of household storage rice had Cd concentrations four times above the permissible Cd level (0.1 mg/kg) [35]. The reported prevalence of low eGFR in Mae Sot was 16.1% [36].

The Institutional Ethical Committees of Chulalongkorn University, Chiang Mai University, and Mae Sot Hospital approved the study protocol [37]. Details of the study objectives, study procedures, benefits, and potential risks were given to all subjects. They were apparently healthy and had resided at their current addresses for 30 years or longer. Exclusion criteria were pregnancy, breastfeeding, a history of metalwork, and a hospital record or physician’s diagnosis of an advanced chronic disease. All subjects gave informed consent prior to participation.

### 2.2. Quantitation of Exposure and Its Effects

Cd exposure and its effects were assessed by a one-time measurement of urinary excretion of Cd, β_2_M, albumin, and NAG. The estimated GFR (eGFR) was calculated using the chronic kidney disease epidemiology collaboration (CKD-EPI) equations [38], which have been validated with inulin clearance [39].

Blood samples were collected within 3 h of urine collection after an overnight fast. Urine, blood, and plasma samples were stored at −80 °C for later analysis for Cd, β_2_M, albumin, NAG, and creatinine.

Graphite furnace atomic absorption spectrometry with the Zeeman effect background correction system was used to quantify Cd in the urine samples. For instrument calibration, multielement standards (Merck KGaA, Darmstadt, Germany) were used. The reference urine metal control levels 1, 2, and 3 (Lyphocheck, Bio-Rad, Hercules, CA, USA) were employed for quality control and quality assurance of Cd quantitation. The coefficient of variation in Cd in the reference urine was within acceptable clinical chemistry standards.

The limit of detection (LOD) for Cd was 0.1 µg/L. This LOD figure was 3 times the standard deviation of a repeated measurement of blank samples. When a urine sample contained Cd below its LOD, the Cd concentration assigned was the LOD value divided by the square root of 2 [40].

Urinary and plasma creatinine concentrations were measured by the colorimetric method, based on the alkaline picrate Jaffe’s reaction [41]. The urinary NAG assay was based on colorimetry (NAG test kit, Shionogi Pharmaceuticals, Sapporo, Japan). The urinary β_2_M assay was based on the latex immunoagglutination method (LX test, Eiken 2MGII; Eiken and Shionogi Co., Tokyo, Japan). The urinary albumin concentration was determined by an immunoturbidimetric method [42,43].

### 2.3. Normalization of Excretion Rate of Cd, β_2_M, alb and NAG

E_x_ was normalized to creatinine clearance (C_cr_) as E_x_/C_cr_ = [Cd]_u_[cr]_p_/[cr]_u_, where x = Cd, β_2_M, alb, or NAG; [x]_u_ = the urine concentration of x (mass/volume); [cr]_p_ = the plasma creatinine concentration (mg/dL); and [cr]_u_ = the urine creatinine concentration (mg/dL). E_x_/C_cr_ was expressed as the amount of x excreted per volume of the glomerular filtrate [26]. This C_cr_-normalization corrects for urine dilution and the number of functioning nephrons, and it is not influenced by muscle mass [44].

E_x_ was normalized to E_cr_ as [x]_u_/[cr]_u_, where x = Cd, β_2_M, alb, or NAG; [x]_u_ = the urine concentration of x (mass/volume); and [cr]_u_ = the urine creatinine concentration (mg/dL). E_x_/E_cr_ was expressed in μg/g creatinine. This E_cr_-normalization corrects for urine dilution only, and it is affected by muscle mass which varies widely among people. Previously, E_cr_-adjustment was found to introduce imprecision or non-differential errors, which prejudiced a dose–response relationship to null [45,46].

### 2.4. Mediation Analysis

Mediation analysis was employed to explore the cause–effect inference of Cd exposure (E_Cd_), tubular injury (E_NAG_), and a reduction in eGFR [47]. Specifically, a simple mediation model, described by MacKinnon et al. [48], was used to test whether Cd decreases eGFR via tubular injury that releases NAG into tubular filtrate and is excreted in urine. The Sobel test, described by Preacher and Hayes [49], was used to determine the statistical significance of the indirect effect of Cd.

Models depicting a tubular injury marker (E_NAG_) as a mediator [M] of an indirect effect of Cd [X] on the dependent variable eGFR [Y] were detailed fully in Section 3.3 together with unstandardized β coefficients and the Sobel test results.

### 2.5. Statistical Analysis

Data were analyzed with IBM SPSS Statistics 21 (IBM Inc., New York, NY, USA). Male–female differences in mean values of continuous variables were assessed with the Mann–Whitney U test. Differences in percentages were assessed with Pearson’s chi-squared test. Conformity to a normal distribution was checked by the Kolmogorov–Smirnov test. Logarithmic transformation was applied to the variables, showing a right-skewed distribution. Histograms showing the distribution of age, eGFR, E_Cd,_ and E_NAG_ are provided in Appendix A.

Predictors of E_NAG_ and eGFR were obtained by multiple linear regression analysis. Determinants of the prevalence odds ratio (POR) for low eGFR and albuminuria were obtained by logistic regression analysis. Quantification of the variability of eGFR (adjusted R^2^), the contribution of Cd, and other variables to the eGFR variability (eta square, η^2^ values) were obtained by univariate analysis with Bonferroni correction in multiple comparisons. For all tests, *p*-values ≤ 0.05 were considered to indicate statistical significance.

## 3. Results

### 3.1. Characteristics of Study Subjects

Environmental Cd exposure levels and demographic characteristics of study subjects can be found in Table 1.

The present cohort consisted of 737 persons with a mean age of 48.1 (range: 16–87 years) and an overall mean E_Cd_/C_cr_ of 0.051 µg/L filtrate and a mean E_Cd_/E_cr_ of 3.72 µg/g creatinine. The overall percentages of women and smokers were 60.7% and 42.7%, respectively. Hypertension occurred more commonly (32.2%) than a low eGFR (9.1%). There were 192 and 545 persons in low- and high-Cd burden groups, defined as an E_Cd_/C_cr_ level below 0.01 and ≥ 0.01 µg/L filtrate, respectively. Smoking was highly prevalent among men in both groups (43.5% vs. 81.8%). Age and eGFR histograms are provided in Appendix A.

In the low-Cd burden group, men and women were in equal numbers and they had the same mean values for all parameters measured, with the exception of age, where the mean age in women was 6.2 years older than men. There were no male–female differences in mean values for E_Cd_/E_cr_, E_NAG_/E_cr_, E_Cd_/C_cr_, or E_NAG_/C_cr_. In the high-Cd burden group, the mean values for E_β2M_/E_cr_ and E_β2M_/C_cr_ were lower in females than males.

In the high Cd-burden group, women constituted 62%, but they were of the same age as the men (a mean age of 52.8 for men and 50.4 for women). The mean BMI, mean eGFR, and % of hypertension all were higher in women than men. However, the mean E_Cd_/E_cr_ in women was lower, compared to men (4.21 vs. 6.01 µg/g of creatinine), most likely due to the very high prevalence of smoking in men (81.8% vs. 32.3%). The mean values for E_NAG_/E_cr_, E_Cd_/C_cr,_ and E_NAG_/C_cr_ in men and women of this high-Cd burden group were similar.

### 3.2. The Risks of a Low eGFR and Albuminuria Increase with the Severity of Tubular Proteinuria

We used logistic model regression to evaluate the association of low eGFR with Cd exposure and other independent variables (Table 2).

As the data in Table 2 indicate, the prevalence odds ratio (POR) values for both a low eGFR and albuminuria rose with increasing age, E_Cd_/C_cr_, and the severity of tubular proteinuria. The risk of a low eGFR, but not albuminuria, was associated with BMI. Per each year increase in age, the respective POR values for low eGFR and albuminuria rose 13.3% and 10%, respectively. The POR for a low eGFR and albuminuria both increased twofold per doubling rE_Cd_/C_cr_. In parallel, POR values for a low eGFR and albuminuria rose with the severity of tubular proteinuria in a dose-dependent manner.

In an equivalent analysis using E_Cd_ and E_β2M_ normalized to E_cr_ (Appendix A), similar results were obtained for age and BMI, while a doubling E_Cd_/E_cr_ was associated with a smaller increase in the POR value for a low eGFR (POR = 1.24), compared with E_Cd_/C_cr_ (POR = 2.09). A significant increase in POR for a low eGFR (POR = 9.7) was found only in those with severe tubular proteinuria (E_β2M_/E_cr_ >1000 µg/g creatinine). The increased risk of having a low eGFR among those with an E_β2M_/E_cr_ rate of 300–1000 µg/g of creatinine did not reach a statistically significant level.

### 3.3. Associations of E_β2M_ with E_Cd_, eGFR, and E_NAG_

Table 3 reports the strength of the association of E_β2M_ with E_Cd_, eGFR, and other independent variables.

As the data in Table 3 indicate, E_β2M_/C_cr_ did not show a significant association with age, hypertension, smoking, or gender, but it was differentially associated with BMI, E_Cd_/C_cr,_ and eGFR. E_β2M_/C_cr_ rates varied directly with E_Cd_/C_cr_ rates in men (β = 0.316), women (β = 0.331), and the high-Cd burden group (β = 0.310) while showing an inverse association with the eGFR in men (β = −0.472), women (β = −0.287), and the high-Cd burden group (β = −0.367). Like the eGFR, the E_β2M_/C_cr_ rate was inversely associated with BMI in men (β = −0.158), women (β = −0.129), and the high-Cd burden group (β = −0.117). E_β2M_/C_cr_ in the low-Cd burden group was not associated with E_Cd_/C_cr_ (β = 0.007) or the eGFR (β = −0.035).

Results obtained from an equivalent analysis using E_β2M_ and E_Cd_ normalized to E_cr_ can be found in Appendix A.

Table 4 reports the strength of the association of E_alb_ with E_Cd_, eGFR, and other independent variables.

E_alb_/C_cr_ did not show a significant association with BMI, hypertension, smoking, or gender, but it was differentially associated with age, E_Cd_/C_cr,_ and eGFR. E_alb_/C_cr_ varied directly with E_Cd_/C_cr_ rates in men (β = 0.250), women (β = 0.169), and the high-Cd burden group (β = 0.210), while showing an inverse association with the eGFR in men (β = −0.460), women (β = −0.309), and the high-Cd burden group (β = −0.351). Seven independent variables altogether did not explain variation in E_alb_/C_cr_ rates in the low-Cd burden group, but they accounted for 41.6%, 33.2%, and 37.6% of the E_alb_/C_cr_ differences among men, women, and the high Cd-burden group, respectively.

Results obtained from an equivalent analysis using E_alb_ and E_Cd_ normalized to E_cr_ can be found in Appendix A.

### 3.4. Moderate-to-Strong Association of E_NAG_/C_cr_ with E_Cd_/C_cr_

Results of the multiple linear regression of the tubular injury, E_NAG_/C_cr,_ can be found in Table 5.

Age, BMI, the eGFR, and smoking appeared to have differential influences on E_NAG_/C_cr_ rates that depended on gender and the body burden level of Cd (Table 3). In comparison, E_Cd_/C_cr_ and hypertension were consistently associated with E_NAG_/C_cr_ rates across four subgroups.

Age showed an inverse association with E_NAG_ in females (β = −0.170) and the high Cd burden group (β = −0.124), but not in males or the low-Cd burden group. Per one-year increases in age, E_NAG_/C_cr_ rates fell to 1.016 × 10^0.007^ (95% CI: 1.016 × 10^0.011^, 1.016 × 10^0.002^) U/L filtrate in females and 1.016 × 10^0.004^ (95% CI: 1.016 × 10^0.08^, 1.016 × 10^0.001^) U/L filtrate in the high-Cd burden group. These were based on the unstandardized β values of log_10_ (E_NAG_/C_cr_) × 10^3^ versus age in females and the high-Cd-burden groups being −0.007 and −0.004, respectively. These data can be interpreted to suggest a decrease in the number of tubular cells or the net loss of tubular cells as age increased, especially among women and those with high-Cd burden after adjustment for potential confounders.

E_NAG_/C_cr_ varied directly with E_Cd_/C_cr_ in men (β = 0.447), women (β = 0.394), the low-Cd burden group (β = 0.287), and the high-Cd burden group (β = 0.145). Similarly, a positive association between E_NAG_/C_cr_ rates and hypertension was observed in men (β = 0.167), women (β = 0.169), the low-Cd burden group (β = 0.180), and the high-Cd burden group (β = 0.158).

E_NAG_/C_cr_ rates varied directly with BMI in only women (β = 0.132), while showing an inverse association with age in women (β = −0.170) and the high-Cd burden group (β = −0.124). An inverse association of E_NAG_/C_cr_ and the eGFR was found also in women (β = −0.178) and the high-Cd burden group (β = −0.223). In comparison, an inverse association of E_NAG_/E_cr_ rates and the eGFR was statistically insignificant in men, women, and the low-Cd burden group (Appendix A).

### 3.5. Quantification of Effects of Cadmium and Tubular Injury on eGFR

The results of logistic regression of the low eGFR are provided in Table 6.

The prevalence odds ratio (POR) for a low eGFR was little affected by gender, hypertension, and smoking, while age, BMI, E_Cd_/C_cr,_ and E_NAG_/C_cr_ quartile 4 levels were associated with increases in the risk of having a low eGFR. Per each year increase in age, and per one kg/m^2^ increase in BMI, the POR for a low eGFR rose 16.7% (*p* = 0.001) and 10.9% (*p* = 0.037), respectively. The POR for a low eGFR increased 2.71-fold per doubling E_Cd_/C_cr_ (*p* < 0.001) and 4.80-fold when E_NAG_/C_cr_ rates rose from quartile 1 to quartile 4 (*p* = 0.015).

Results of a univariate analysis of the eGFR can be found in Table 7.

More than half (63.3%) of male eGFR variation was accounted for (Table 7). Age contributed the largest fraction (34%), followed by E_Cd_/C_cr_ rates (8.1%) and hypertension (1.6%). In women, 44% of their eGFR variation was accounted for. Age, E_Cd_/C_cr_, and E_NAG_/C_cr_ contributed 24.9%,11.4%, and 3.4% of the variation, respectively.

In the low-Cd burden group, 49.4% of the total eGFR variation was accounted for. Age contributed the most to eGFR variability (38%), followed by smoking × hypertension × E_NAG_/C_cr_ interactions (5.9%) and gender (5.1%). E_Cd_/C_cr_ contributed to only 0.034% of eGFR variation (*p* = 0.828).

In the high-Cd burden group, seven independent variables together accounted fornearly half of the total variation in the eGFR (49.3%) with no interaction complications. Age contributed the most to eGFR variability (30%) followed by E_Cd_/C_cr_ (15%) and E_NAG_/C_cr_ quartiles (1.3%).

### 3.6. A Cause-Effect Inference Analysis of Cd Exposure, Tubular Injury (E_NAG_), and GFR Decline

The scatterplots of the variables in mediation analysis for the low-Cd burden group are provided in Figure 1.

A linear dose–response relationship was evident for log [(E_NAG_/C_cr_) × 10^3^] vs. log [(E_Cd_/C_cr_) × 10^5^] (Figure 1a), eGFR vs. log [(E_Cd_/C_cr_) × 10^5^] (Figure 1b) in both men and women. For the eGFR vs. log [(E_NAG_/C_cr_) × 10^3^], a dose–response relationship was present only in men (Figure 1c).

The mediation analysis model for the low-Cd burden group is provided in Figure 2.

In a simple mediation analysis model (Figure 2a), there was a significant effect of E_Cd_/C_cr_ on the eGFR (β = −0.374, *p* < 0.001). However, its effect was not mediated through E_NAG_/C_cr_, as suggested by a nonstatistical significance figure of *a*b* (*p* = 0.258) (Figure 2b).

The scatterplots of the variables in mediation for the high-Cd burden group are provided in Figure 3.

A linear dose–response relationship was evident for log [(E_NAG_/C_cr_) × 10^3^] vs. log [(E_Cd_/C_cr_) × 10^5^] (Figure 3a), the eGFR vs. log [(E_Cd_/C_cr_) × 10^5^] (Figure 3b), and the eGFR vs. log [(E_NAG_/C_cr_) × 10^3^] (Figure 1c) in both men and women.

The results of the mediation analysis model for the high-Cd burden group are provided in Figure 4.

In a simple mediation model (Figure 4a), there was a significant effect of E_Cd_/C_cr_ on the eGFR (β = −0.482, *p* < 0.001). However, its effect was not mediated through E_NAG_/C_cr_ as the Sobel test informed a nonstatistical significance figure of *a*b* (*p* = 0.139) (Figure 4b).

## 4. Discussion

The present study recruited 737 participants from a low-exposure location (Bangkok) and the Mae Sot district, Tak province, where Cd contamination was endemic. Based on their E_Cd_ levels, 192 subjects had E_Cd_/C_cr_ < 0.01 µg/L of filtrate (a geometric mean of 0.41 µg/g of creatinine) and they were assigned to the low-Cd burden group (Table 1). The remaining 545 subjects were assigned to the high-Cd burden group (a geometric mean of 5.45 µg/g of creatinine). None of the low-Cd burden group had a low eGFR. This was expected, given that subjects in the low0Cd burden group were young (a mean age of 35.8 for men and 42.1 for women), and therefore their renal Cd accumulation had not yet reached a toxic level. In other Cd research studies, those aged 50 years or older were recruited to maximize the likelihood of finding kidney effects. Such study groups were not representative of the general population. The present study recruited 737 people aged 16 to 87 years, and their age histograms indicated they could be considered as representative of the general population (Appendix A).

The limitations of the present study are acknowledged. They include a one-time-only assessment of Cd exposure and its effects and the relatively small numbers of subjects in the low-Cd burden group (*n* = 192). A population-based design, involving the recruitment of participants from their communities precluded renal pathological assessment by renal biopsy. The strengths of the present study were the assessment of adverse outcomes on tubular and glomerular functions of the kidneys in subjects with a wide range of environmental Cd exposure (Appendix A).

### 4.1. Use of β_2_-Microglobulinuria in Toxicological Risk Assessment of Cadmium Exposure

In following JECFA’s toxicological risk assessment of Cd in the human diet [12], most studies assumed E_β2M_/E_cr_ ≥ 300 µg/g of creatinine as an adverse health effect of concern. Compelling evidence, however, suggests that a E_β2M_/E_cr_ rate above 300 µg/g of creatinine is a predictor of serious adverse health effects such as hypertension [50] and a fall of the eGFR at high rates [51,52]. Consistent with these observations are data showing that E_β2M_ was inversely associated with the eGFR (Table 3), while the risks of having a low eGFR and albuminuria increased with the severity of β_2_-microglobulinuria (Table 2 and Appendix A). The POR for a low eGFR rose 3.2-fold and 16.2-fold as E_β2M_/C_cr_ rates increased to 3–10 and >10 µg/L of filtrate, respectively. In parallel, the POR for albuminuria rose 2.3-fold and 3.9-fold (Table 2). These data question the use of the β_2_M endpoint as a criterion to judge the nephrotoxicity of Cd, and to calculate a safe dietary Cd exposure limit. In theory, the most sensitive endpoint should be used as a basis from which exposure limits are determined [6,53].

### 4.2. Cadmium Body Burden at Which Adverse Kidney Outcomes Were Observed

Cd-induced tubular injury, assessed with an increased E_NAG_, has often been observed, especially in environmental low-dose exposure scenarios. For example, in a study from the United Kingdom (U.K.), an E_Cd_/E_cr_ rate of 0.5 μg/g of creatinine was associated with 2.6- and 3.6-fold increases in the likelihood of having abnormal NAG excretion (E_NAG_/E_cr_ > 2 U/g of creatinine), compared to an E_Cd_/E_cr_ rate of 0.3 and <0.5 μg/g of creatinine, respectively [54]. Hence, a significant increase in risk of having kidney damage has been linked to an E_Cd_/E_cr_ rate as low as 0.5 μg/g of creatinine. This E_Cd_/E_cr_ rate was one-tenth of the current threshold at 5.24 μg/g of creatinine.

At least 30 publications have shown a dose–response relationship of E_NAG_/E_cr_ and E_Cd_/E_cr_ [55]. Like urine Cd, urine NAG emanates from injured or dying tubular cells [13,26,27], such that an excreted amount of NAG is proportional to the number of surviving nephrons, and E_NAG_ and E_Cd_ can be expected to be closely correlated as shown previously [13]. For these reasons, the present study focused on E_NAG_ together with the eGFR, which is a criterion used in clinical trials to evaluate CKD treatment outcomes [3,4,5]. This research was conducted to explore if E_NAG_ could be causally linked to a declining eGFR, using a simple mediation analysis (Figure 2 and Figure 4).

E_Cd_ and E_NAG_ are most logically normalized to a function of intact nephron mass because they both are released by tubular cells [6]. The GFR is the measurable analog of the nephron number; if C_cr_ is accepted as a surrogate for the GFR, C_cr_-normalization corrects for differences in nephron mass. C_cr_ normalization may overstate the toxicity implied by a robust E_Cd_ when the nephron number is normal, and may understate the toxicity implied by a modest E_Cd_ when the nephron number is reduced. The impact of E_cr_-normalization of E_Cd_ and E_NAG_ is illustrated in the SM (Appendix A).

Through multiple regression analysis (Table 5), E_NAG_/C_cr_ was strongly associated with E_Cd_/C_cr_ in men (β = 0.447) and women (β = 0.394). A linear dose–response relationship in every subgroup was indicated clearly by scatterplots (Figure 1a and Figure 3a). Similarly, an inverse association of the eGFR and E_Cd_/C_cr_ was consistent across subgroups (Figure 1b and Figure 3b). Notably, however, E_NAG_/C_cr_ was inversely associated with the eGFR in women (β = −0.178) and the high-Cd burden group (β = −0.223), but not in men or the low Cd-burden group (Table 2). These discrepancies could be interpreted to suggest that a Cd-induced reduction in the eGFR and Cd-induced tubular injury, which caused NAG release, were independent or causally unrelated.

The results of the mediation models lend support to the above interpretation; a significant effect of E_Cd_/C_cr_ on the eGFR was suggested for both the low- (β = −0.374) and the high-Cd burden groups (β = −0.482) (Figure 2a and Figure 4a), while the Sobel test results informed nonstatistical significance figures of a mediation effect (*a*b*) in both groups (Figure 2b and Figure 4b). Therefore, Cd-induced injury that causes the release of NAG appeared to play little or no role in the nephron destruction that reduces the GFR. The molecular basis of Cd-induced nephron destruction was not apparent from the present study.

As data in Table 6 and Table 7 indicate, the risk of having a low eGFR was influenced by age, BMI, the level of body burden of Cd, and the severity of tubular injury. A high-Cd body burden and tubular injury contributed, respectively, to 15% (*p* < 0.001) and 1.3% (*p* = 0.085) of the variation in the eGFR among those who had E_Cd_/C_cr_ ≥ 0.01 µg/L of filtrate. In comparison, gender and a low-Cd body burden contributed, respectively, to 5.1% (*p* = 0.007) and 0.034% (*p* = 0.828) in the eGFR variation among subjects who had E_Cd_/C_cr_ < 0.01 µg/L of filtrate. These findings suggest that Cd exposure level producing an E_Cd_/C_cr_ rate below 0.01 µg/L of filtrate was the least likely to induce significant damage to kidneys. Similar results were obtained from the multiple regression modeling of E_β2M_ and E_alb_ (Table 3 and Table 4).

An E_Cd_/C_cr_ rate of less than 0.01 µg/L of filtrate is in the ranges with the benchmark dose limit (BMDL) of Cd body burden [46]. At present, the BMDL is a replacement to the no-observed-adverse-effect level (NOAEL) due to its shortcoming. The NOAEL is referred to as the highest experimental dose level for which the response does not significantly differ from the response in the control group [53]. Using the E_NAG_/E_cr_ and eGFR data from 790 Swedish women aged 53–64 years, the BMD figures of E_Cd_/E_cr_ were 0.5–0.8 and 0.7–1.2 μg/g of creatinine, respectively [56]. From this Swedish study, it can be inferred that “safe” Cd exposure levels include those producing an E_Cd_/E_cr_ rate of less than 0.5 μg/g of creatinine. A U.K. study [54], however, suggested some lower E_Cd_/E_cr_ figures (<0.3 μg/g of creatinine).

An increased risk of CKD can now be attributable to dietary exposure [14,57]. In the China Health and Nutrition Survey (*n* = 8429), of which 641 (7.6%) of participants had CKD, the likelihood of having CKD rose 1.73-fold, 2.93-fold, and 4.05-fold in those with respective Cd intake amounts of 23.2, 29.6, and 36.9 μg/day, compared to a Cd intake amount of 16.7 μg/day [57]. From this Chinese study, an inferred “safe” dietary Cd exposure level would be below 16.7 µg/day. This level is 28.8% of the JECFA tolerable intake figure of Cd at 58 µg/day [12].

### 4.3. The Tubulo-Glomerular Effects of Cadmium

Consistent with the literature on Cd toxicity, tubular cell injury (E_NAG_) and defective tubular re-absorptive functions (E_β2M_) both were Cd-dose dependent. E_NAG_ and E_β2M_ both were inversely related to the eGFR since GFR reduction is a common sequela of ischemic acute tubular necrosis and acute and chronic tubulointerstitial fibrosis [4,5,58]. Indeed, primary glomerular disease also leads to tubulointerstitial inflammation and fibrosis, presumably because reabsorbed proteins are toxic to tubular cells [59,60]. Whether glomeruli or tubules are injured initially, the extent of tubulointerstitial fibrosis correlates best with the GFR in CKD [61,62,63], given that acute tubular necrosis and acute and chronic tubulointerstitial fibrosis all create impediments to filtration such as the destruction of post-glomerular peritubular capillaries, the amputation of glomeruli from tubules, and the obstruction of nephrons with cellular debris.

By simple mediation analysis, it was found that the reduction in the GFR due to Cd was not causally related to the tubular release of NAG by Cd (Figure 2 and Figure 4). The renal tubules’ capacity to regenerate and repair mild-to-moderate injury [64,65] may explain such observation. At a high dose level, Cd may, however, limit the kidneys’ capacity to repair injured tubules (Table 4). The GFR begins, consequently, to fall as tubular cell damage and death are intensified with continuing Cd influx, and tubular proteinuria ensues because of defective reabsorption and the loss of functioning nephrons (Table 2 and Table 6).

Intriguingly, E_β2M_ was independently associated with an increased risk of having hypertension in a Japanese population study [50]. In a recent study using mediation analysis, hypertension in Cd-exposed people was a consequential result of GFR reduction by Cd [66]. Also, Cd may impose premature tubular cell senescence [67,68,69], which limits further the repair and regeneration of tubular cells, resulting in a net loss of tubular cells, evident from a decrease in E_NAG_ per nephron as age increased when confounders were adjusted (Table 5). More likely, the presented data suggested that the mechanisms of Cd-induced nephron destruction that reduces the GFR are different from those causing the tubular release of NAG.

### 4.4. Cadmium-Induced Microalbuminuria

CKD is now the seventh most common cause of death from non-communicable disease worldwide, and its incidence is projected to increase further as its major risk factors, obesity, diabetes, hypertension, and non-alcoholic fatty liver continue to rise [1,2]. Urinary albumin could be of utility in screening CKD in its early stage [3,4,5]. In a recent human study, Cd has been shown to reduce the fractional reabsorption of both albumin and β_2_M by the same extent (18–21%) [70]. In rats, Cd caused albuminuria by dampening the cubilin/megalin receptor system, which involved in protein reabsorption [71]. In cultured LLC-PK1 cells and pig proximal tubular cells, Cd reduced albumin reabsorption through lowering the expression levels of megalin and chloride channel 5 (ClC5) [72]. Increased glomerular permeability to albumin was observed in the other studies using human renal glomerular endothelial cells in monolayers, where a non-cytotoxic concentration of Cd (1 µM) caused the redistribution of the adherens junction proteins, vascular endothelial–cadherin, and β-catenin [73,74].

The pathogenesis of Cd-induced albuminuria, especially in low environmental Cd exposure conditions (E_Cd_/E_cr_ < 0.5 µg/g creatinine), requires further study, especially in experimentation given that the glomerular and tubular causes of albuminuria may not be distinguishable in epidemiological studies.

## 5. Conclusions

The risks of having a low eGFR and albuminuria rose, respectively, 2.7-fold and 2.0-fold per twofold increase in the body burden of Cd. Severe tubular injury increased the risk of a low eGFR by a further 43.8%. Mediation analysis inferred that the destruction of nephrons by Cd, which reduces the GFR, occurs via mechanisms other than those causing NAG release. Environmental Cd exposure that produces an E_Cd_/C_cr_ rate of less than 0.01 µg/L of filtrate contributes to 0.034% of the eGFR variability. Thus, an E_Cd_/C_cr_ rate of 0.01 µg/L filtrate could be used to reliably define the body burden of Cd, below which there is no excess risk of kidney damage. This Cd exposure level is in ranges with the NOAEL equivalent obtained by benchmark dose computation.

## Figures and Tables

**Figure 1 toxics-12-00775-f001:**
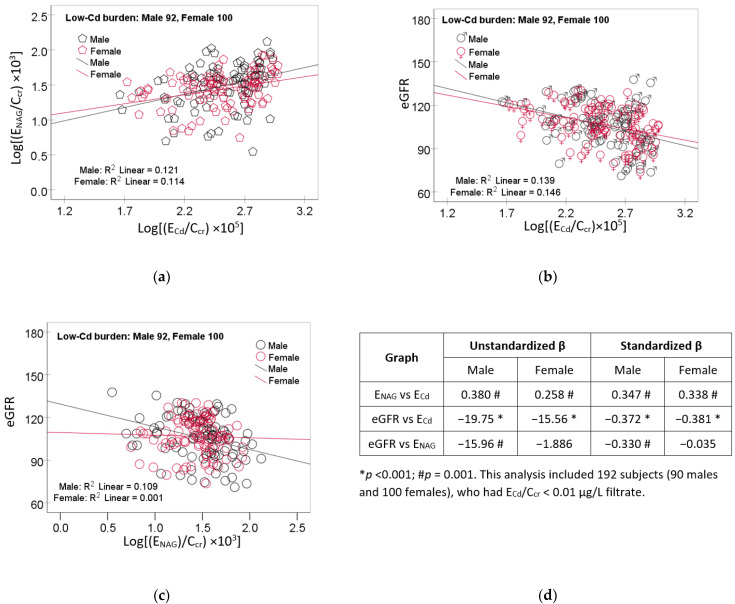
Linear dose–response relationships of NAG, the eGFR, and cadmium excretion in the low-Cd burden group. Scatterplots relate log [(E_NAG_/C_cr_) × 10^3^] to log [(E_Cd_/C_cr_) × 10^5^] (**a**), eGFR to log [(E_Cd_/C_cr_) × 10^5^] (**b**), and eGFR to log [(E_NAG_/C_cr_) × 10^3^] (**c**) in men and women who had E_Cd_/C_cr_ < 0.01 µg/L of filtrate. Coefficients of determination (R^2^) and *p*-values are provided in each graph. Table (**d**) provides unstandardized and standardized β values and *p*-values.

**Figure 2 toxics-12-00775-f002:**
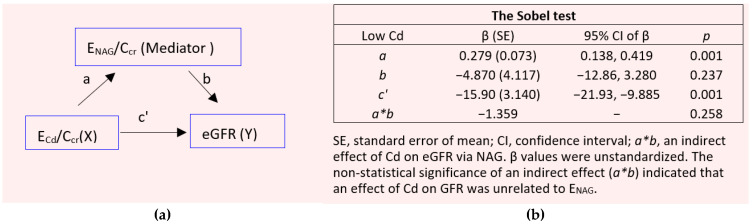
Analysis of tubular injury as a mediator of the cadmium effect on GFR in the low-Cd burden group. The model depicts E_NAG_/C_cr_ as a mediator of the effect of Cd on the eGFR (**a**) and the Sobel test (**b**) of unstandardized β coefficients describing relationships of E_Cd_/C_cr_ with ENAG/Ccr (*a*), E_NAG_/C_cr_ with eGFR (*b*), and E_Cd_/C_cr_ with the eGFR (*c’*).

**Figure 3 toxics-12-00775-f003:**
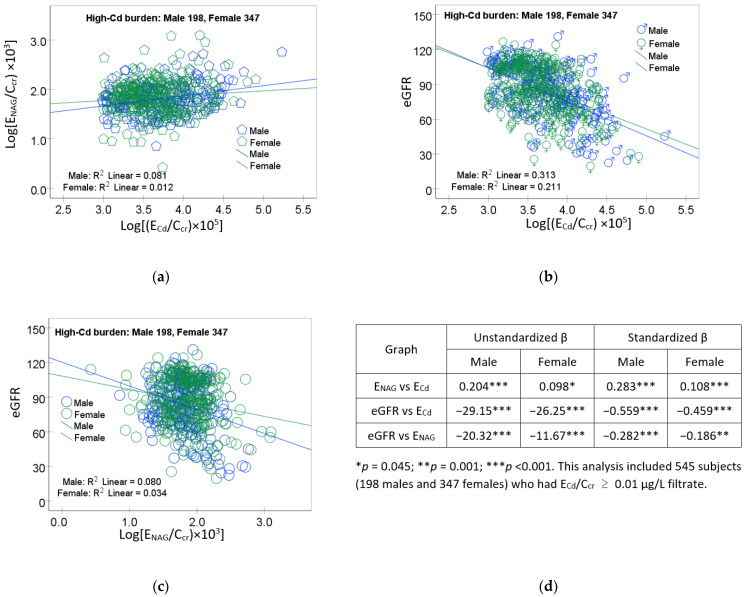
Linear dose–response relationships of NAG, eGFR, and cadmium excretion in the high-Cd burden group. Scatterplots relate log [(E_NAG_/C_cr_) × 10^3^] to log [(E_Cd_/C_cr_) × 10^5^] (**a**), the eGFR to log [(E_Cd_/C_cr_) × 10^5^] (**b**), and eGFR to log [(E_NAG_/C_cr_) × 10^3^] (**c**) in men and women. Coefficients of determination (R^2^) and *p*-values are provided in each graph. Table (**d**) provides unstandardized and standardized β values and *p*-values.

**Figure 4 toxics-12-00775-f004:**
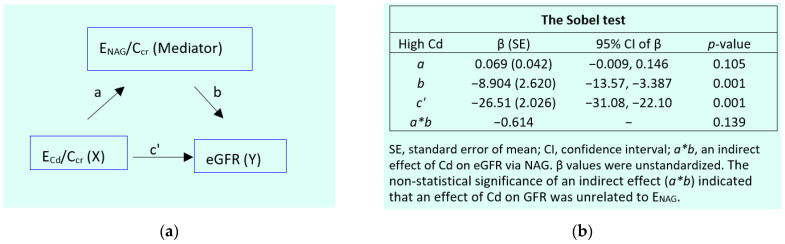
An analysis of tubular injury as a mediator of the cadmium effect on the GFR in the high-Cd burden group. The model depicts E_NAG_/C_cr_ as a mediator of the effect of Cd on the eGFR (**a**) and the Sobel test (**b**) of unstandardized β coefficients describing relationships of E_Cd_/C_cr_ and E_NAG_/C_cr_ (*a*), E_NAG_/C_cr_ with eGFR (*b*), and E_Cd_/C_cr_ with the eGFR (*c’*).

**Table 1 toxics-12-00775-t001:** Descriptive characteristics of participants according to gender and cadmium burden levels.

Parameters	All, *n* = 737	Low-Cd Burden ^a^		High-Cd Burden	
		Males, *n* = 92	Females, *n* = 100	Males, *n* = 198	Females, *n* = 347
Age, years	48.1 (11.0)	35.8 (10.2)	42.1 (9.1) ***	52.8 (11.3)	50.4 (8.2)
Age range, years	16–87	16–87	23–60	30–87	33–84
BMI, kg/m^2^	23.2 (3.8)	23.3 (3.4)	23.4 (3.8)	22.0 (3.3)	23.7 (4.1) ***
% Female	60.7	−	52.0	−	63.7
% Smoking	42.7	43.5	1.0 ***	81.8	32.3 ***
% Hypertension	32.2	27.0	15.8	30.9	39.2 *
eGFR ^a^, mL/min/1.73 m^2^	91 (22)	106 (16)	107 (13)	83 (22)	87 (21) *
% eGFR ≤ 60 mL/min/1.73 m^2^	9.1	0	0	15.2	10.7
[cr]_p_, mg/dL	0.88 (0.28)	0.92 (0.12)	0.67 (0.10) ***	1.08 (0.34)	0.82 (0.23) ***
[cr]_u_, mg/dL	110.2 (73.8)	89.8 (80.3)	70.3 (58.3)	135.6 (65.2)	112.6 (74.5) ***
[Cd]_u_, µg/L	6.53 (11.71)	0.39 (0.48)	0.44 (0.57)	10.7 (18.7)	7.52 (7.82)
Normalized to E_cr_ (E_x_/E_cr_) ^b^					
E_Cd_/E_cr_, µg/g creatinine	2.78 (0.60)	0.25 (0.29)	0.28 (0.31)	4.72 (0.38)	5.17 (0.33) *
E_NAG_/E_cr_, units/g creatinine	2.40 (1.41)	3.12 (0.32)	3.41 (0.32)	2.37 (0.44)	2.03 (0.40)
E_alb_/E_cr_, mg/g creatinine	5.19 (0.71)	4/34 (0.52)	3.98 (0.27)	4.89 (0.75)	5.41 (0.70)
E_β2M_/E_cr_, µg/g creatinine	18.38 (1.27)	6.7 (1.04)	6.91 (1.42)	47.19 (1.42)	18.58 (1.20) **
% E_β2M_/E_cr_, µg/g creatinine					
<300	88.1	95.7	99	76.9	89.0
300–1000	6.3	3.3	1.0	11.3	6.0
>1000	5.6	1.1	0	11.8	5.1
Normalized to C_cr_, (E_x_/C_cr_) ^c^					
(E_Cd_/C_cr_) × 100, µg/L filtrate	2.36 (0.63)	0.32 (0.29)	0.32 (0.32)	5.27 (0.43)	4.50 (0.36)
(E_NAG_/C_cr_) × 100, µg/L filtrate	5.43 (0.35)	3.07 (0.32)	2.73 (0.24)	6.56 (0.31)	6.92 (0.33)
(E_alb_/C_cr_) × 100, mg/L filtrate	11.28 (1.02)	4.05 (0.51)	2.80 (0.28)	17.89 (1.08)	9.12 (0.97) **
E_β2M_/C_cr_ × 100, µg/L filtrate	41.53 (1.20)	6.59 (1.17)	5.5 (1.05)	130.62 (1.20)	63.01 (1.00) **
% E_β2M_/C_cr_ × 100, µg/L filtrate					
<300	82.0	95.7	100	66.2	82.3
300–1000	8.0	3.3	0	14.6	7.8
>1000	10.00	2.1	0	19.2	9.9

*n*, number of subjects; BMI, body mass index; eGFR, estimated glomerular filtration rate; cr, creatinine; alb, albumin; Cd, cadmium. ^a^ Low and high burdens of Cd were defined as E_Cd_/C_cr_ < 0.01 and ≥ 0.01 µg/L filtrate. eGFR was determined using the CKD-EPI equations. ^b^ E_Cd_/E_cr_ = [Cd]_u_/[cr]_u_. ^c^ E_Cd_/C_cr_ = [Cd]_u_[cr]_p_/[cr]_u_. Data for BMI, Eβ_2_M, and E_alb_ were from 712, 735, and 531 subjects, respectively. All other data were from 737 subjects. E_Cd_, E_β2M_, E_alb_ and E_NAG_ data are presented as geometric mean and geometric standard deviation (SD) values. All other continuous variables are expressed as arithmetic mean and arithmetic SD values. For all tests, *p* ≤ 0.05 identifies statistical significance, determined with the Pearson chi-squared test for differences in percentages and the Mann–Whitney U for male–female differences in mean values. *** *p* < 0.001, ** *p* = 0.001–0.004, and * *p* = 0.034–0.038.

**Table 2 toxics-12-00775-t002:** The prevalence odds ratios for low eGFR and albuminuria in relation to the severity of tubular proteinuria assessed by β_2_M excretion normalized to creatinine clearance.

Independent Variables/Factors	Low eGFR ^a^		Albuminuria ^b^	
	POR (95% CI)	*p*	POR (95% CI)	*p*
Age, years	1.133 (1.084, 1.183)	<0.001	1.100 (1.064, 1.137)	<0.001
BMI, kg/m^2^	1.156 (1.039, 1.287)	0.008	0.957 (0.896, 1.023)	0.197
Log_2_[(E_Cd_/C_cr_) × 10^5^], µg/L filtrate	2.093 (1.512, 2.898)	<0.001	1.977 (1.597, 2449)	<0.001
Gender	0.475 (0.202, 1.118)	0.088	1.299 (0.771, 2.188)	0.326
Hypertension	2.127 (0.939, 4.816)	0.070	0.654 (0.408, 1.048)	0.078
Smoking	1.371 (0.585, 3.212)	0.467	1.059 (0.641, 1.749)	0.823
E_β2M_/C_cr_, µg/L filtrate				
<3	Referent		Referent	
3–10	3.227 (1.111, 9.373)	0.031	2.334 (1.100, 4.955)	0.027
>10	16.20 (6.432, 20.78)	<0.001	3.904 (1.501, 10.16)	0.005

POR, prevalence odds ratio; CI, confidence interval; β, regression coefficient; BMI, body mass index; eGFR, estimated glomerular filtration rate. ^a^ Low eGFR was defined as eGFR ≤ 60 mL/min/1.73 m^2^. ^b^ Albuminuria was defined as E_alb_/C_cr_ ≥ 0.2 mg/L filtrate in men and women. Numbers of subjects in groups having E_β2M_/C_cr_ < 3, 3–10, and >10 µg/L filtrate were 572, 58, and 72, respectively. For all tests, *p*-values ≤ 0.05 indicate statistical significance.

**Table 3 toxics-12-00775-t003:** Predictors of the excretion of β_2_M normalized to creatinine clearance.

Independent Variables/ Factors	Log_10_[(E_β2M_/C_cr_) × 10^3^], µg/L Filtrate							
	Males, *n* = 277		Females, *n* = 425		Low-Cd Burden ^a^, *n* = 166		High-Cd Burden, *n* = 536	
	β	*p*	β	*p*	β	*p*	β	*p*
Age, years	−0.101	0.171	0.037	0.472	0.082	0.512	0.051	0.241
BMI, kg/m^2^	−0.158	0.001	−0.129	0.002	−0.121	0.149	−0.117	0.001
Log_2_[(E_Cd_/C_cr_) × 10^5^], µg/L filtrate	0.316	<0.001	0.331	<0.001	0.007	0.942	0.310	<0.001
eGFR, mL/min/1.73 m^2^	−0.472	<0.001	−0.287	<0.001	−0.035	0.752	−0.367	<0.001
Hypertension	−0.030	0.509	0.000282	0.994	0.016	0.853	0.001	0.981
Smoking	0.052	0.259	0.069	0.099	0.137	0.132	0.035	0.358
Gender	−	−	−	−	0.045	0.648	−0.037	0.311
Adjusted R^2^	0.483	<0.001	0.383	<0.001	−0.012	0.655	0.448	<0.001

*n*, number of subjects; eGFR, estimated glomerular filtration rate; β, standardized regression coefficient; BMI, body mass index; adjusted R^2^, coefficient of determination. ^a^ Low and high levels of Cd burden were indicated by E_Cd_/C_cr_ < 0.01 and ≥0.01 µg/L filtrate. β indicates the strength of the association of E_β2M_/C_cr_ with seven independent variables (first column). The adjusted R^2^ indicates the proportion of the variation of rates of E_β2M_/C_cr_, which was explained by all independent variables. *p*-values ≤ 0.05 indicate statistically significant associations of independent variables with E_β2M_/C_cr_.

**Table 4 toxics-12-00775-t004:** Predictors of albumin excretion rate normalized to creatinine clearance.

Independent Variables/ Factors	Log_10_[(E_alb_/C_cr_) × 10^4^], mg/L Filtrate							
	Males, *n* = 277		Females, *n* = 425		Low-Cd Burden ^a^, *n* = 166		High-Cd Burden, *n* = 536	
	β	*p*	β	*p*	β	*p*	β	*p*
Age, years	0.020	0.810	0.205	<0.001	0.263	0.724	0.136	0.004
BMI, kg/m^2^	−0.034	0.593	−0.071	0.159	−0.349	0.512	−0.059	0.145
Log_2_[(E_Cd_/C_cr_) × 10^5^], µg/L filtrate	0.250	<0.001	0.169	0.001	0.373	0.566	0.210	<0.001
eGFR, mL/min/1.73 m^2^	−0.460	<0.001	−0.309	<0.001	0.099	0.836	−0.351	<0.001
Hypertension	−0.018	0.769	−0.024	0.602	0.871	0.190	−0.017	0.640
Smoking	0.018	0.759	0.029	0.546	−0.420	0.416	0.042	0.309
Gender	−	−	−	−	−0.614	0.321	−0.024	0.546
Adjusted R^2^	0.416	<0.001	0.332	<0.001	−0.103	0.607	0.376	<0.001

*n*, number of subjects; eGFR, estimated glomerular filtration rate; β, standardized regression coefficient; BMI, body mass index; adjusted R^2^, coefficient of determination. ^a^ Low and high levels of Cd burden were indicated by E_Cd_/C_cr_ < 0.01 and ≥0.01 µg/L filtrate. β indicates the strength of the association of E_alb_/C_cr_ with seven independent variables (first column). The adjusted R^2^ indicates the proportion of the variation of rates of E_alb_/C_cr_, which was explained by all independent variables. *p*-values ≤ 0.05 indicate statistically significant associations of independent variables with E_alb_/C_cr_.

**Table 5 toxics-12-00775-t005:** Predictors of tubular injury, measured as E_NAG_/C_cr_.

Independent Variables/ Factors	Log_10_[(E_NAG_/C_cr_) × 10^3^], U/L Filtrate							
	Males, *n* = 277		Females, *n* = 427		Low-Cd Burden ^a^, *n* = 186		High-Cd Burden, *n* = 538	
	β	*p*	β	*p*	β	*p*	β	*p*
Age, years	−0.012	0.892	−0.170	0.003	−0.094	0.419	−0.124	0.026
BMI, kg/m^2^	0.002	0.974	0.132	0.003	−0.008	0.924	0.087	0.063
Log_2_[(E_Cd_/C_cr_) × 10^5^], µg/L filtrate	0.447	<0.001	0.394	<0.001	0.287	0.001	0.145	0.004
eGFR, mL/min/1.73 m^2^	−0.127	0.127	−0.178	0.002	−0.132	0.205	−0.223	<0.001
Hypertension	0.167	0.002	0.169	<0.001	0.180	0.022	0.158	<0.001
Smoking	−0.037	0.496	0.111	0.016	−0.055	0.517	0.045	0.356
Gender	−	−	−	−	−0.061	0.511	0.045	0.343
Adjusted R^2^	0.293	<0.001	0.266	<0.001	0.114	<0.001	0.090	<0.001

*n*, number of subjects; eGFR, estimated glomerular filtration rate; β, standardized regression coefficient; BMI, body mass index; adjusted R^2^, coefficient of determination. ^a^ Low and high levels of Cd burden were indicated by E_Cd_/C_cr_ < 0.01 and ≥0.01 µg/L filtrate. β indicates the strength of the association of E_NAG_/C_cr_ with seven independent variables (first column). The adjusted R^2^ indicates the proportion of the variation of rates of E_NAG_/C_cr_, which was explained by all independent variables. *p*-values ≤ 0.05 indicate statistically significant associations of independent variables with E_NAG_/C_cr_.

**Table 6 toxics-12-00775-t006:** Determinants of the prevalence odds ratios for low eGFR.

Independent Variables/Factors	Low eGFR				
	β Coefficients	POR	95% CI		*p*
	(SE)		Lower	Upper	
Age, years	0.156 (0.022)	1.168	1.118	1.221	<0.001
BMI, kg/m^2^	0.104 (0.050)	1.109	1.006	1.222	0.037
Log_2_[(E_Cd_/C_cr_) × 10^5^], µg/L filtrate	0.998 (0.164)	2.714	1.967	3.744	<0.001
Gender	−0.389 (0.429)	0.678	0.292	1.572	0.365
Hypertension	−0.707 (0.387)	0.493	0.231	1.052	0.068
Smoking	0.268 (0.414)	1.307	0.581	2.945	0.517
Tubular injury ^a^					
Minimal	Referent				
Mild	−0.529 (0.754)	0.589	0.134	2.581	0.483
Moderate	0.218 (0.694)	1.244	0.320	4.843	0.753
Severe	1.570 (0.648)	4.804	1.350	17.09	0.015

POR, prevalence odds ratio; CI, confidence interval; β, regression coefficient; SE, standard error of the mean; BMI, body mass index; eGFR, estimated glomerular filtration rate. ^a^ Minimal, mild, moderate, and severe tubular injury were defined according to E_NAG_/C_cr_ quartiles 1, 2, 3, and 4, respectively. Arithmetic means (SD) of (E_NAG_/C_cr_) × 100 in the minimal, mild, moderate, and severe tubular groups were 2.19 (0.78), 4.29 (0.62), 6.90 (0.96), and 17.02 (14.96) U/L filtrate, respectively. Corresponding numbers of subjects in the Cd burden groups were 190, 171, 172, and 171. For all tests, *p*-values ≤ 0.05 indicate statistical significance.

**Table 7 toxics-12-00775-t007:** Predictors of eGFR reductions and the magnitude of their effects.

Independent Variables/ Factors	eGFR, mL/min/1.73 m^2^							
	Males, *n* = 277		Females, *n* = 427		Low-Cd Burden ^a^, *n* = 186		High-Cd Burden, *n* = 538	
	η^2^	*p*	η^2^	*p*	η^2^	*p*	η^2^	*p*
Age, years	0.340	<0.001	0.249	<0.001	0.380	<0.001	0.300	<0.001
BMI, kg/m^2^	0.001	0.637	0.004	0.228	0.002	0.632	0.010	0.024
Log_2_[(E_Cd_/C_cr_) × 10^5^], µg/L filtrate	0.081	<0.001	0.114	<0.001	0.000335	0.828	0.150	<0.001
Smoking	0.002	0.434	0.000171	0.792	0.000407	0.811	0.000248	0.724
Hypertension	0.016	0.044	0.002	0.420	0.000012	0.968	0.004	0.167
E_NAG_/C_cr_ quartiles	0.015	0.275	0.034	0.003	0.018	0.460	0.013	0.085
Gender	−	−	−	−	0.051	0.007	0.003	0.256
Smoking × Hypertension	−	−	0.022	0.003	0.013	0.176	0.004	0.147
Smoking × Hypertension × E_NAG_/C_cr_ quartile	−	−	−	−	0.059	0.014	−	−
Adjusted R^2^	0.633	<0.001	0.440	<0.001	0.494	<0.001	0.493	<0.001

*n*, number of subjects; eGFR, estimated glomerular filtration rate; η^2^, eta squared; BMI, body mass index; adjusted R^2^, coefficient of determination. ^a^ Low and high burdens of Cd were indicated by E_Cd_/C_cr_ < 0.01 and ≥0.01 µg/L filtrate. η^2^ indicates a proportion of eGFR variability explained by each independent variable (first column). The adjusted R^2^ indicates the proportion of the variation of E_NAG_/C_cr_, which was explained by all independent variables. *p*-values ≤ 0.05 indicate statistical significance.

## Data Availability

All data are contained within this article.

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
