# Peer review of "Urinary N-acetylglucosaminidase in People Environmentally Exposed to Cadmium Is Minimally Related to Cadmium-Induced Nephron Destruction"

_toxics, 2024, doi:10.3390/toxics12110775_

Round 1
Reviewer 1 Report (Previous Reviewer 1)
Comments and Suggestions for Authors
As mentioned in the previous review comments, Cd can not only induce degeneration or apoptosis of renal tubular epithelial cells, but also lead to glomerular atrophy, inflammatory cell infiltration and oxidative stress damage in renal tissue. Therefore, the decrease in eGFR caused by Cd is related to multiple factors, and several biomarkers are needed to more accurately reflect the renal injury caused by Cd. In the revised manuscript, the authors added β2-microglobulin as a biomarker to evaluate the renal tubular injury caused by Cd. I suggest the ratio of urinary microalbumin to creatinine need to supplement in the study to reflect the glomeruli injury. If possible, it is best to include an assessment of renal pathological damage.
Author Response
Reviewer 1
Comments and Suggestions
As mentioned in the previous review comments, Cd can not only induce degeneration or apoptosis of renal tubular epithelial cells, but also lead to glomerular atrophy, inflammatory cell infiltration and oxidative stress damage in renal tissue. Therefore, the decrease in eGFR caused by Cd is related to multiple factors, and several biomarkers are needed to more accurately reflect the renal injury caused by Cd. In the revised manuscript, the authors added β2-microglobulin as a biomarker to evaluate the renal tubular injury caused by Cd. I suggest the ratio of urinary microalbumin to creatinine need to supplement in the study to reflect the glomeruli injury. If possible, it is best to include an assessment of renal pathological damage.
RESPONSES:
- Thank you for your advice to further improve my manuscript.
- Accordingly, I have now provided data on albumin excretion and albuminuria prevalence.
- Results of logistic regression analysis of albuminuria and predictors of albumin excretion rate have been inserted into Table 2 (lines 191- ) and new Table 4 (lines 232-250).
- In effect, data in Table 2 show that Cd increased similarly the risk of having both low eGFR and albuminuria.
- Part of the Introduction has been rewritten to better reflect the study objectives and to include the current understanding of Eβ2M and Ealb (lines 59-76), quoted below.
- New results on albuminuria are discussed in subheading 4.4 Cadmium and microalbuminuria (lines 484-501), quoted below.
- Assessment of renal pathologies (by renal biopsy) was not possible because the present study was a population-based design, involving recruitment participants from their communities. This has been declared as one of the study limitations (lines 374-380).
Ample evidence suggests that exposure to low-concentrations of Cd increases the risks of a low eGFR and albuminuria [14, 25], but research into the mechanisms by which Cd induces these pathologies are scarce. The present study has two major aims. First aim, is to explore the potential cause-effect inference of Cd exposure level, tubular injury, and a declining eGFR through simple mediation analysis. Second aim, is to determine the body burden of Cd at which these outcomes occur, using data on urinary excretion of β2-microglobulin (β2M), albumin and N-acetyl-β-D-glucosaminidase (NAG) as kidney effect biomarkers. Urinary NAG emanates from injured or dying tubular cells, and thus its excretion reflects kidney injury from any causes [26,27].
The protein β2M is filtered freely by the glomeruli and is reabsorbed almost completely by the kidney’s tubular cells [28, 29]. Cd has been shown to cause a reduction in the tubular reabsorption of β2M [30], and increased β2M excretion has been used as an indicator of tubulopathy. In comparison, the protein albumin is not normally filtered by glomeruli, due to its large molecular weight, and its negative charge [31-33]. By means of transcytosis through endothelial cells and podocyte foot processes, albumin reaches the proximal tubular lumen [34], and is reabsorbed and returned to the circulation by three major mechanisms; fluid phase endocytosis, the megalin/cubillin receptor-mediated endocytosis and transcytosis [31-33].
[26] Price, R.G. Measurement of N-acetyl-beta-glucosaminidase and its isoenzymes in urine: Methods and clinical applications. Eur. J. Clin. Chem. Clin. Biochem. 1992, 30, 693–705.
[27] Pócsi, I.; Dockrell, M.E.; Price, R.G. Nephrotoxic biomarkers with specific indications for metallic pollutants: Implications for environmental health. Biomark. Insights 2022, 17, 11772719221111882.
[28] Argyropoulos, C.P.; Chen, S.S.; Ng, Y.H.; Roumelioti, M.E.; Shaffi, K.; Singh, P.P.; Tzamaloukas, A.H. Rediscovering beta-2 microglobulin as a biomarker across the spectrum of kidney diseases. Front. Med. 2017, 4, 73.
[29] Portman, R.J.; Kissane, J.M.; Robson, A.M. Use of B2-microglobulin to diagnose tubulo-interstitial renal lesions in children. Kidney Int. 1986, 30, 91–98.
[30] Gauthier, C.; Nguyen-Simonnet, H.; Vincent, C.; Revillard, J.-P.; Pellet, M.V. Renal tubular absorption of beta 2 micro-globulin. Kidney Int. 1984, 26, 170–175.
[31] Nielsen, R.; Christensen, E.I.; Birn, H. Megalin and cubilin in proximal tubule protein reabsorption: From experimental models to human disease. Kidney Int. 2016, 89, 58–67.
[32] Molitoris, B.A.; Sandoval, R.M.; Yadav, S.P.S.; Wagner, M.C. Albumin uptake and processing by the proximal tubule: Physiological, pathological, and therapeutic implications. Physiol. Rev. 2022, 102, 1625–1667.
[33] Comper, W.D.; Vuchkova, J.; McCarthy, K.J. New insights into proteinuria/albuminuria. Front. Physiol. 2022, 13, 991756.
[34] Benzing, T.; Salant, D. Insights into glomerular filtration and albuminuria. N. Engl. J. Med. 2021, 384, 1437–1446.
4.4 Cadmium-induced microalbuminuria
CKD is now the seventh most common cause of death from non-communicable disease worldwide, and its incidence is projected to increase further as its major risk factors, obesity, diabetes, hypertension, and non-alcoholic fatty liver continue to rise [1,2]. Urinary albumin could be of utility in screening CKD in its early stage [3-5]. In a recent human study, Cd has been shown to reduce the fractional reabsorption of both albumin and β2M by the same extent (18-21%) [70]. In rats, Cd caused albuminuria by dampening the cubilin/megalin receptor system, [71]. In cultured LLC-PK1 cells and pig proximal tubular cells, Cd reduced albumin reabsorption through lowering expression levels of megalin and chloride channel 5 (ClC5) [72]. An increased glomerular permeability to albumin was observed in the other studies, using human renal glomerular endothelial cells in monolayers, where a non-cytotoxic concentration of Cd (1 µM) caused the redistribution of the adherens junction proteins, vascular endothelial-cadherin and β-catenin [73,74].
The pathogenesis of Cd-induced albuminuria, especially in low environmental Cd exposure conditions, requires further study, especially in experimentation, given that the glomerular and tubular causes of albuminuria may not be distinguishable in epidemiologic studies.
Reviewer 2 Report (Previous Reviewer 2)
Comments and Suggestions for Authors
Minor Comments
1. Do not use abbreviations (GFR) in the title.
2. Table 1: For variables that are log-transformed in later calculations, please indicate geometric mean and geometric standard deviation instead. (Cd, Cr, NAG, β2M)
3. Table 2: please use Log2-transformation instead of Log10 for (ENAG/CCr), to be consistent with Table 3-5.
4. Table 3-4: For ease of interpretation, e.g. β of 0.007 should be expressed as 1.016 times (100.007) and those 95% confidence intervals of ‘times’ should be added.
5. Fig1,2, S3~5: Please indicate like ‘log10 ’ instead of just ‘log’ to indicate the base used.
Author Response
Reviewer 2
Thank you for additional comments and suggestions to improve a manuscript. Accordingly, I have undertaken all tasks the reviewers offered.
Minor Comments
Point 1. Do not use abbreviations (GFR) in the title.
RESPONSE: GFR in the title has been spelt out.
Point 2. Table 1: For variables that are log-transformed in later calculations, please indicate geometric mean and geometric standard deviation instead. (Cd, Cr, NAG, β2M)
RESPONSE: The geometric means and geometric standard deviations have now been given for the continuous variables that were subjected to logarithmic transformation. These included data on urinary excretion of Cd, NAG, albumin and β2M.
Point 3. Table 2: please use Log2-transformation instead of Log10 for (ENAG/CCr), to be consistent with Table 3-5. (ECd/Ccr log 2)
RESPONSE: A correction has been made as advised.
Point 4. Table 3-4: For ease of interpretation, e.g. β of 0.007 should be expressed as 1.016 times
(10 0.007) and those 95% confidence intervals of ‘times’ should be added.
RESPONSE: Thank you very much for this helpful suggestion. Corrections have now been undertaken.
Point 5. Fig1,2, S3~5: Please indicate like ‘log10 ’ instead of just ‘log’ to indicate the base used.
RESPONSE: Base 10 logarithm has been specified where required in SM Figures.
Reviewer 3 Report (New Reviewer)
Comments and Suggestions for Authors
The work presented for review is a valuable study demonstrating the need for research toxic substances, including toxic elements, especially those present in the diet and showing toxic effects at low exposure levels of the general population. The author of this paper focused on the role of NAG, beta2 microglobulin release in nephron destruction that reduces eGFR in Cd-induced injury. However, the research model used, the selected parameters of renal injury and the methods used require some clarification. Main objections: - The author emphasizes in the Discussion that the study focuses on ENAG together with eGFR, which is a criterion used in clinical trials to evaluate CKD treatment outcomes (lines: 354-355). Why didn't the author perform any other tests related to renal function, such as the assessment of albumin or proteinuria in general or also ACR (albumin-to-creatinine ratio) and uric acid. The lack of additional studies detracts from the role of determining NAG itself, which is a non-specific marker and can also be increased in diabetes and hypertension, as the author examined, but also with therapy with aminoglycosides, valproate and methotrexate or cisplatin. In addition, urinary NAG levels are increased in poisoning with other metals, i.e. lead or mercury. This issue should be discussed or added as a limitation of the study - Study group selection - the author pays special attention to the role of Cd exposure at low environmental doses (<0.5 ug/g creatinine) while the study group used mostly (n=545) consists of “high-Cd burden” subjects and n=192 with “low-Cd burden,” which reduces the usefulness of the analysis precisely for those subjects with lower Cd levels. In addition, the selection of individuals for the study groups also raises some concerns, since according to Table 2, there is not a single woman and not a single man in the low-Cd burden group with eGFR < 60 ml/min/1.73 m2. These issues would need to be discussed. - Why were the results in Table 2 statistically analyzed by gender only? The author of the paper emphasizes in the introduction as the purpose of the paper the performance of a dose-response relationship and a cause-effect inference of Cd exposure. It would also be valuable to demonstrate a comparative analysis for groups characterized by different Cd levels, mainly in terms of the proposed normalization of Cd results and comparison with the hitherto accepted reporting of urinary Cd concentration result as ug/g creatinine - method of determining beta2 microglobulin is missing. Add sample collection, sample preparation and basic validation parameters Minor objections: - Standardize the notation of GFR (line 23) and eGFR - Please provide the test number from the manufacturer of the NAG determination and the basic parameters: CV, sensitivity and LOD/LOQ - Please provide details of Cd determination in urine or literature including description of atomic absorption spectrometry used and validation parameters of the method (sensitivity, precision, recovery, LOD, LOQ, range), as well as description of sample preparation.
Comments on the Quality of English Language
Minor editing of English language required.
Author Response
Reviewer 3
Comments and Suggestions
The work presented for review is a valuable study demonstrating the need for research toxic substances, including toxic elements, especially those present in the diet and showing toxic effects at low exposure levels of the general population. The author of this paper focused on the role of NAG, beta2 microglobulin release in nephron destruction that reduces eGFR in Cd-induced injury. However, the research model used, the selected parameters of renal injury and the methods used require some clarification.
RESPONSE:
- Thank you for reviewing my work, insightful comments, and helpful suggestions to improve a manuscript. Accordingly, I have undertaken necessary revisions detailed below for the issues raised. Changes to the text are in blue.
- I have rewritten part of the Introduction to clarify study objectives and the renal biomarkers selection (lines 57-74), quoted below.
Ample evidence suggests that exposure to low-concentrations of Cd increases the risks of a low eGFR and albuminuria [14, 25], but research into the mechanisms by which Cd induces these pathologies are scarce. The present study has two major aims. First aim, is to explore the potential cause-effect inference of Cd exposure level, tubular injury, and a declining eGFR through simple mediation analysis. Second aim, is to determine the body burden of Cd at which these outcomes occur, using data on urinary excretion of β2-microglobulin (β2M), albumin and N-acetyl-β-D-glucosaminidase (NAG) as kidney effect biomarkers. Urinary NAG emanates from injured or dying tubular cells, and thus its excretion reflects kidney injury from any causes [26,27].
The protein β2M is filtered freely by the glomeruli and is reabsorbed almost completely by the kidney’s tubular cells [28, 29]. Cd has been shown to cause a reduction in the tubular reabsorption of β2M [30], and increased β2M excretion has been used as an indicator of tubulopathy. In comparison, the protein albumin is not normally filtered by glomeruli, due to its large molecular weight, and its negative charge [31-33]. By means of transcytosis through endothelial cells and podocyte foot processes, albumin reaches the proximal tubular lumen [34], and is reabsorbed and returned to the circulation by three major mechanisms; fluid phase endocytosis, the megalin/cubillin receptor-mediated endocytosis and transcytosis [31-33].
Main objections
Point 1: The author emphasizes in the Discussion that the study focuses on ENAG together with eGFR, which is a criterion used in clinical trials to evaluate CKD treatment outcomes (lines: 354-355). Why didn't the author perform any other tests related to renal function, such as the assessment of albumin or proteinuria in general or also ACR (albumin-to-creatinine ratio) and uric acid. The lack of additional studies detracts from the role of determining NAG itself, which is a non-specific marker and can also be increased in diabetes and hypertension, as the author examined, but also with therapy with aminoglycosides, valproate and methotrexate or cisplatin. In addition, urinary NAG levels are increased in poisoning with other metals, i.e. lead or mercury. This issue should be discussed or added as a limitation of the study.
RESPONSE: Thank you for raising this non-specific nature of urinary NAG. As advised, I have now included albumin excretion and albuminuria as additional renal parameters. These are clinically relevant parameters.
- Results of logistic regression analysis of albuminuria and predictors of albumin excretion rate have been inserted into Table 2 (lines 191- ) and new Table 4 (lines 232-250).
- In effect, data in Table 2 show that Cd increased similarly the risk of having both low eGFR and albuminuria.
- I have now provided an overview of a study in the first paragraph of the Discussion and the limitations (lines).
Point 2: Study group selection - the author pays special attention to the role of Cd exposure at low environmental doses (<0.5 ug/g creatinine) while the study group used mostly (n=545) consists of “high-Cd burden” subjects and n=192 with “low-Cd burden,” which reduces the usefulness of the analysis precisely for those subjects with lower Cd levels. In addition, the selection of individuals for the study groups also raises some concerns, since according to Table 2, there is not a single woman and not a single man in the low-Cd burden group with eGFR < 60 ml/min/1.73 m2. These issues would need to be discussed. - Why were the results in Table 2 statistically analyzed by gender only?
RESPONSE:
- I agree with the reviewer that a small number of subjects (n = 192) who had low Cd-burden could be considered as a limitation, and it has now been acknowledged (lines 374-380).
- I disagree with the reviewer that a lack of subjects with low eGFR in the low-Cd burden was a concern (Table 1). This was not unexpected, given that subjects in the low burden group were younger (mean age 8 for men and 42.1 for women), and their kidney Cd burden has not yet reached a toxic level. Most Cd research toxicity studies included those aged 50 years or older, which were not representative of the general population. The present study included 737 persons, aged 19 to 87 years, and their age histogram indicated they could be considered as representative of the general population (please see Figure 1S).
- In Table 2, where analysis of a low eGFR prevalence included all subjects, and ECd/Ccr was entered as a continuous variable, after adjustment for age, BMI and other potential confounders, the risk of having low eGFR rose twofold per doubling of ECd/Ccr and the severity of tubular proteinuria. These results are strengthened by analysis of albuminuria prevalence that the reviewer suggested.
- New results on albuminuria are discussed in subheading 4.4 Cadmium and microalbuminuria (lines 484-501), quoted below.
4.4 Cadmium-induced microalbuminuria
CKD is now the seventh most common cause of death from non-communicable disease worldwide, and its incidence is projected to increase further as its major risk factors, obesity, diabetes, hypertension, and non-alcoholic fatty liver continue to rise [1,2]. Urinary albumin could be of utility in screening CKD in its early stage [3-5]. In a recent human study, Cd has been shown to reduce the fractional reabsorption of both albumin and β2M by the same extent (18-21%) [70]. In rats, Cd caused albuminuria by dampening the cubilin/megalin receptor system, [71]. In cultured LLC-PK1 cells and pig proximal tubular cells, Cd reduced albumin reabsorption through lowering expression levels of megalin and chloride channel 5 (ClC5) [72]. An increased glomerular permeability to albumin was observed in the other studies, using human renal glomerular endothelial cells in monolayers, where a non-cytotoxic concentration of Cd (1 µM) caused the redistribution of the adherens junction proteins, vascular endothelial-cadherin and β-catenin [73,74].
The pathogenesis of Cd-induced albuminuria, especially in low environmental Cd exposure conditions, requires further study, especially in experimentation, given that the glomerular and tubular causes of albuminuria may not be distinguishable in epidemiologic studies.
Point 2.1: The author of the paper emphasizes in the introduction as the purpose of the paper the performance of a dose-response relationship and a cause-effect inference of Cd exposure. It would also be valuable to demonstrate a comparative analysis for groups characterized by different Cd levels, mainly in terms of the proposed normalization of Cd results and comparison with the hitherto accepted reporting of urinary Cd concentration result as ug/g creatinine - method of determining beta2 microglobulin is missing. Add sample collection, sample preparation and basic validation parameters.
RESPONSE:
- The referred objective statement on a dose-response relationship analysis was in error and has been deleted. My intention was to analyze the interrelationships of Cd exposure and its effects using a simple mediation analysis (Figures 1 and Figure 2).
- Results of dose-response analyses comparing Ccr- and Ecr-normalizations and the prevalence of low eGFR across ECd/Ccr groups, BMI groups, and age-groups (as categorial variables have been published (please see ref. ).
- The missing method for determining urinary β2-mmicroglobulin has been now provided together with a method for urinary albumin assay.
[] Satarug, S.; Đorđević, A.B.; Yimthiang, S.; Vesey, D.A.; Gobe, G.C. The NOAEL Equivalent of Environmental Cadmium Exposure Associated with GFR Reduction and Chronic Kidney Disease. Toxics 2022, 10, 614.
Minor objections: - Standardize the notation of GFR (line 23) and eGFR - Please provide the test number from the manufacturer of the NAG determination and the basic parameters: CV, sensitivity and LOD/LOQ - Please provide details of Cd determination in urine or literature including description of atomic absorption spectrometry used and validation parameters of the method (sensitivity, precision, recovery, LOD, LOQ, range), as well as description of sample preparation.
RESONSES:
- I have checked the proper use of the terms GFR and eGFR in this manuscript.
- As it is stated in the method section, data analyzed in the present study were taken from previously published reports, where readers can find full information regarding analytical methods. It would be redundant to duplicate the methodology details.

Round 2
Reviewer 3 Report (New Reviewer)
Comments and Suggestions for Authors
All suggestions were taken into account and corrected.
This manuscript is a resubmission of an earlier submission. The following is a list of the peer review reports and author responses from that submission.
Round 1
Reviewer 1 Report
Comments and Suggestions for Authors
It‘s interested that the author elaborated in detail on the correlation between renal cadmium (Cd) excretion and the concentration of NAG in the urine, and evaluated the reduced glomerular filtration function depend on Cd induced injury of renal tubular cells. However, the progressive decline in eGFR reflects long-term and sustained renal function damage owing to glomerular and tubulointerstitial injury, rather than short-term and recoverable changes in renal function. The strong regenerative ability of renal tubular epithelial cells can self repair and renew after injury to some extent. Persistent damage to renal tubular function only occurs when the degree of damage exceeds the repair capacity. Thus, the concentration of Cd is not directly proportional to the degree of renal tubular injury. In addition, Cd can not only induce degeneration or apoptosis of renal tubular epithelial cells, but also lead to glomerular atrophy, inflammatory cell infiltration and oxidative stress damage in renal tissue. Therefore, the decrease in eGFR caused by Cd is related to multiple factors, and several biomarkers are needed to more accurately reflect the renal injury caused by Cd. I suggest it could not be published before more meaningful results of biomarkers associated with the mechanisms of Cd-induced kidney injury are supplemented.
Author Response
Reviewer 1
Comments and Suggestions
It‘s interested that the author elaborated in detail on the correlation between renal cadmium (Cd) excretion and the concentration of NAG in the urine, and evaluated the reduced glomerular filtration function depend on Cd induced injury of renal tubular cells. However, the progressive decline in eGFR reflects long-term and sustained renal function damage owing to glomerular and tubulointerstitial injury, rather than short-term and recoverable changes in renal function. The strong regenerative ability of renal tubular epithelial cells can self repair and renew after injury to some extent. Persistent damage to renal tubular function only occurs when the degree of damage exceeds the repair capacity. Thus, the concentration of Cd is not directly proportional to the degree of renal tubular injury. In addition, Cd can not only induce degeneration or apoptosis of renal tubular epithelial cells, but also lead to glomerular atrophy, inflammatory cell infiltration and oxidative stress damage in renal tissue. Therefore, the decrease in eGFR caused by Cd is related to multiple factors, and several biomarkers are needed to more accurately reflect the renal injury caused by Cd. I suggest it could not be published before more meaningful results of biomarkers associated with the mechanisms of Cd-induced kidney injury are supplemented.
RESPONSE:
I thank the reviewer for evaluating my work, insightful comments, suggestions, and guidance to improve a paper. The essential revisions have been undertaken to address the critical issues raised by the reviewer. Changes to the text are in blue and are listed below.
- Additional results and conclusion are inserted to the abstract (lines 14-17 and lines 23-25)
- An analysis of another marker of tubular effect, namely Eβ2M has been inserted in the study objective (lines 60-67).
- New Tables 2 and 3 have been inserted to report findings from an additional analysis of Eβ2M normalized to Ccr (lines 178-220)
- New Tables S1 and S2 have been added to the SM to provide results from analysis of Eβ2M normalized to Ecr.
- The interpretations of results from Eβ2M analysis are provided in the Discussion (lines 324-332).
- New paragraphs have been constructed, quoted below, to explain thoroughly and meaningfully the findings (lines 402-427).
- Fifteen more references have been added.
Consistent with the literature on Cd toxicity, tubular injury (ENAG) and tubular malfunction (Eβ2M) both were Cd-dose dependent. ENAG and Eβ2M both were inversely related to eGFR since GFR reduction is a common sequela of ischemic acute tubular necrosis and acute and chronic tubulointerstitial fibrosis [4,5,49]. Indeed, primary glomerular disease also leads to tubulointerstitial inflammation and fibrosis, presumably because reabsorbed proteins are toxic to tubular cells [50,51]. Whether glomeruli or tubules are injured initially, the extent of tubulointerstitial fibrosis correlates best with GFR in CKD [52-54], given that acute tubular necrosis and acute and chronic tubulointerstitial fibrosis all create impediments to filtration such as the destruction of post-glomerular peritubular capillaries, amputation of glomeruli from tubules, and obstruction of nephrons with cellular debris.
By simple mediation analysis, it was found that the reduction in GFR due to Cd was not causally related to tubular release of NAG by Cd (Figures 2 and 4). The renal tubules’ capacity to regenerate and repair mild-to-moderate injury [55,56], may explain such observation. At a high-dose level, Cd may, however, limit the kidneys’ capacity to repair injured tubules (Table 4). The GFR begins, consequently, to fall as tubular cell damage and death are intensified with continuing Cd influx, and tubular proteinuria ensues because of defective reabsorption and loss of functioning nephrons [Tables 2 and 5].
Intriguingly, Eβ2M was independently associated with risk of hypertension in a Japanese population study [39]. Hypertension was attributable GFR reductions induced by exposure to environmental Cd in a recent study using mediation analysis [57]. Taken together the presented data suggested that Cd-induced nephron destruction that reduces GFR and the tubular release of NAG by Cd involved different mechanisms and kinetics. Also, Cd may induce premature tubular cell senescence [58-60], which limits further the repair and regeneration of tubular cells, resulting in net loss, evident from a decrease in ENAG per nephron with increasing age and adjustment for confounders (Table 4).
[39] Mashima, Y.; Konta, T.; Kudo, K.; Takasaki, S.; Ichikawa, K.; Suzuki, K.; Shibata, Y.; Watanabe, T.; Kato, T.; Kawata, S.; et al. Increases in urinary albumin and beta2-microglobulin are independently associated with blood pressure in the Japanese general population: The Takahata Study. Hypertens. Res. 2011, 34, 831-835.
[40] Kudo, K.; Konta, T.; Mashima, Y.; Ichikawa, K.; Takasaki, S.; Ikeda, A.; Hoshikawa, M.; Suzuki, K.; Shibata, Y.; Watanabe, T.; et al. The association between renal tubular damage and rapid renal deterioration in the Japanese population: The Takahata study. Clin. Exp. Nephrol. 2011, 15, 235-241.
[41] Satarug, S.; Vesey, D.A.; Nishijo, M.; Ruangyuttikarn, W.; Gobe, G.C. The inverse association of glomerular function and urinary β2-MG excretion and its implications for cadmium health risk assessment. Environ. Res. 2019, 173, 40-47.
[49] Lang, S.M.; Schiffl, H. Smoking status, cadmium, and chronic kidney disease. Ren. Replace. Ther. 2024, 10, 17.
[50] Nath, K.A. Tubulointerstitial changes as a major determinant in the progression of renal damage. Am. J. Kidney Dis. 1992, 20, 1-17.
[51] Sharma, S.; Smyth, B. From proteinuria to fibrosis: An update on pathophysiology and treatment options. Kidney Blood Press. Res. 2021, 46, 411-420.
[52] Risdon, R.A.; Sloper, J.C.; De Wardener, H.E. Relationship between renal function and histological changes found in renal-biopsy specimens from patients with persistent glomerular nephritis. Lancet 1968, 292, 363-366.
[53] Schainuck, L.I.; Striker, G.E.; Cutler, R.E.; Benditt, E.P. Structural-functional correlations in renal disease: Part II: The correlations. Human Pathol. 1970, 1, 631-641.
[54] Bohle, A.; von Gise, H.; Mackensen-Haen, S.; Stark-Jakob, B. The obliteration of the postglomerular capillaries and its influence upon the function of both glomeruli and tubuli. Functional interpretation of morphologic findings. Klin. Wochenschr. 1981, 59, 1043-1051.
[55] Kazeminia, S.; Eirin, A. Role of mitochondria in endogenous renal repair. Clin. Sci. (Lond) 2024, 138, 963-973.
[56] Kramann, R.; Kusaba, T.; Humphreys, B.D. Who regenerates the kidney tubule? Nephrol. Dial. Transplant 2015, 30, 903-910.
[57] Satarug, S.; Vesey, D.A.; Yimthiang, S.; Khamphaya, T.; Pouyfung, P.; Đorđević, A.B. Environmental Cadmium Exposure Induces an Increase in Systolic Blood Pressure by Its Effect on GFR. Stresses 2024, 4, 436-451.
[58] Zhang, Y.; Liu, M.; Xie, R. Associations between cadmium exposure and whole-body aging: mediation analysis in the NHANES. BMC Public Health 2023, 23, 1675.
[59] Dong, W.; Zhang, K.; Gong, Z.; Luo, T.; Li, J.; Wang, X.; Zou, H.; Song, R; Zhu, J.; Ma, Y.; et al. N-acetylcysteine delayed cadmium-induced chronic kidney injury by activating the sirtuin 1-P53 signaling pathway. Chem Biol Interact. 2023, 369, 110299.
[60] Chou, X.; Li, X.; Min, Z.; Ding, F.; Ma, K.; Shen, Y.; Sun, D.; Wu, Q. Sirtuin-1 attenuates cadmium-induced renal cell senescence through p53 deacetylation. Ecotoxicol. Environ. Saf. 2022, 245, 114098.
Reviewer 2 Report
Comments and Suggestions for Authors
The study investigated whether tubular damage affects the estimated glomerular filtration rate as a health effect of cadmium. The conclusion is that changes in NAG do not affect glomerular damage, but the non-significant results alone do not indicate such a conclusion to be drawn. This manuscript is therefore not worthy of acceptance for publication.
Major comments
1. Table 1: In the High-Cd burden group, it is difficult to assume SD because the values in brackets are larger than the mean; it needs to be clearly stated whether it is GSD or other.
2. Table 2: For Log[(ENAG/Ccr)×103], U/L filtrate, it should be clarified whether the base was 10 or 2 or other. Also, in the results, it should be stated how the results are interpreted, e.g. how many times the NAG increases with increasing age by one year.
3. Line 186: Table 3→Table 2?
4. Lines 306-308: It is possible that there is insufficient power to detect non-significant results in this study, and the interpretation that there is little or no association is an oversimplification.
5. Lines 313-315: This statement is not appropriate as the variation in NAG in this study cannot be solely attributed to Cd expoaure.
6. Lines 323-324: ‘ECd/Ecr of 0.01-0.02 µg/g creatinine’is too low. It indicates normally undetectable levels, which may indicate a conversion error.
7. Lines 340-341: This is not a valid statement, as no arguments have been presented in this manuscript to deny the present evaluation that urinary Cd is good indicator of Cd body burden.
Author Response
Reviewer 2
Comments and Suggestions
The study investigated whether tubular damage affects the estimated glomerular filtration rate as a health effect of cadmium. The conclusion is that changes in NAG do not affect glomerular damage, but the non-significant results alone do not indicate such a conclusion to be drawn. This manuscript is therefore not worthy of acceptance for publication.
RESPONSE:
Thank you for a thorough evaluation of my work, and offering helpful comments and suggestions for a necessary improvement. Accordingly, I revised a manuscript extensively to fully address all issues and concerns raised by the reviewer. Changes to the text are in blue. Important revisions are listed below.
- An analysis of another marker of tubular effect, namely Eβ2M has been inserted in the study objective (lines 60-67).
- New Tables 2 and 3 have been inserted to report findings from an additional analysis of Eβ2M normalized to Ccr (lines 178-220)
- New Tables S1 and S2 have been added to the SM to provide results from analysis of Eβ2M normalized to Ecr.
- The interpretations of results from Eβ2M analysis are provided in the Discussion (lines 324-332).
- Additional results and conclusion are inserted to the abstract (lines 14-17 and lines 23-25)
- New paragraphs have been constructed, quoted below, to explain thoroughly and meaningfully the findings (lines 402-427).
- Fifteen more references have been added.
Consistent with the literature on Cd toxicity, tubular injury (ENAG) and tubular malfunction (Eβ2M) both were Cd-dose dependent. ENAG and Eβ2M both were inversely related to eGFR since GFR reduction is a common sequela of ischemic acute tubular necrosis and acute and chronic tubulointerstitial fibrosis [4,5,49]. Indeed, primary glomerular disease also leads to tubulointerstitial inflammation and fibrosis, presumably because reabsorbed proteins are toxic to tubular cells [50,51]. Whether glomeruli or tubules are injured initially, the extent of tubulointerstitial fibrosis correlates best with GFR in CKD [52-54], given that acute tubular necrosis and acute and chronic tubulointerstitial fibrosis all create impediments to filtration such as the destruction of post-glomerular peritubular capillaries, amputation of glomeruli from tubules, and obstruction of nephrons with cellular debris.
By simple mediation analysis, it was found that the reduction in GFR due to Cd was not causally related to tubular release of NAG by Cd (Figures 2 and 4). The renal tubules’ capacity to regenerate and repair mild-to-moderate injury [55,56], may explain such observation. At a high-dose level, Cd may, however, limit the kidneys’ capacity to repair injured tubules (Table 4). The GFR begins, consequently, to fall as tubular cell damage and death are intensified with continuing Cd influx, and tubular proteinuria ensues because of defective reabsorption and loss of functioning nephrons [Tables 2 and 5].
Intriguingly, Eβ2M was independently associated with risk of hypertension in a Japanese population study [39]. Hypertension was attributable GFR reductions induced by exposure to environmental Cd in a recent study using mediation analysis [57]. Taken together the presented data suggested that Cd-induced nephron destruction that reduces GFR and the tubular release of NAG by Cd involved different mechanisms and kinetics. Also, Cd may induce premature tubular cell senescence [58-60], which limits further the repair and regeneration of tubular cells, resulting in net loss, evident from a decrease in ENAG per nephron with increasing age and adjustment for confounders (Table 4).
[39] Mashima, Y.; Konta, T.; Kudo, K.; Takasaki, S.; Ichikawa, K.; Suzuki, K.; Shibata, Y.; Watanabe, T.; Kato, T.; Kawata, S.; et al. Increases in urinary albumin and beta2-microglobulin are independently associated with blood pressure in the Japanese general population: The Takahata Study. Hypertens. Res. 2011, 34, 831-835.
[40] Kudo, K.; Konta, T.; Mashima, Y.; Ichikawa, K.; Takasaki, S.; Ikeda, A.; Hoshikawa, M.; Suzuki, K.; Shibata, Y.; Watanabe, T.; et al. The association between renal tubular damage and rapid renal deterioration in the Japanese population: The Takahata study. Clin. Exp. Nephrol. 2011, 15, 235-241.
[41] Satarug, S.; Vesey, D.A.; Nishijo, M.; Ruangyuttikarn, W.; Gobe, G.C. The inverse association of glomerular function and urinary β2-MG excretion and its implications for cadmium health risk assessment. Environ. Res. 2019, 173, 40-47.
[49] Lang, S.M.; Schiffl, H. Smoking status, cadmium, and chronic kidney disease. Ren. Replace. Ther. 2024, 10, 17.
[50] Nath, K.A. Tubulointerstitial changes as a major determinant in the progression of renal damage. Am. J. Kidney Dis. 1992, 20, 1-17.
[51] Sharma, S.; Smyth, B. From proteinuria to fibrosis: An update on pathophysiology and treatment options. Kidney Blood Press. Res. 2021, 46, 411-420.
[52] Risdon, R.A.; Sloper, J.C.; De Wardener, H.E. Relationship between renal function and histological changes found in renal-biopsy specimens from patients with persistent glomerular nephritis. Lancet 1968, 292, 363-366.
[53] Schainuck, L.I.; Striker, G.E.; Cutler, R.E.; Benditt, E.P. Structural-functional correlations in renal disease: Part II: The correlations. Human Pathol. 1970, 1, 631-641.
[54] Bohle, A.; von Gise, H.; Mackensen-Haen, S.; Stark-Jakob, B. The obliteration of the postglomerular capillaries and its influence upon the function of both glomeruli and tubuli. Functional interpretation of morphologic findings. Klin. Wochenschr. 1981, 59, 1043-1051.
[55] Kazeminia, S.; Eirin, A. Role of mitochondria in endogenous renal repair. Clin. Sci. (Lond) 2024, 138, 963-973.
[56] Kramann, R.; Kusaba, T.; Humphreys, B.D. Who regenerates the kidney tubule? Nephrol. Dial. Transplant 2015, 30, 903-910.
[57] Satarug, S.; Vesey, D.A.; Yimthiang, S.; Khamphaya, T.; Pouyfung, P.; Đorđević, A.B. Environmental Cadmium Exposure Induces an Increase in Systolic Blood Pressure by Its Effect on GFR. Stresses 2024, 4, 436-451.
[58] Zhang, Y.; Liu, M.; Xie, R. Associations between cadmium exposure and whole-body aging: mediation analysis in the NHANES. BMC Public Health 2023, 23, 1675.
[59] Dong, W.; Zhang, K.; Gong, Z.; Luo, T.; Li, J.; Wang, X.; Zou, H.; Song, R; Zhu, J.; Ma, Y.; et al. N-acetylcysteine delayed cadmium-induced chronic kidney injury by activating the sirtuin 1-P53 signaling pathway. Chem Biol Interact. 2023, 369, 110299.
[60] Chou, X.; Li, X.; Min, Z.; Ding, F.; Ma, K.; Shen, Y.; Sun, D.; Wu, Q. Sirtuin-1 attenuates cadmium-induced renal cell senescence through p53 deacetylation. Ecotoxicol. Environ. Saf. 2022, 245, 114098.
Major comments
- Table 1: In the High-Cd burden group, it is difficult to assume SD because the values in brackets are larger than the mean; it needs to be clearly stated whether it is GSD or other.
RESPONSE:
- I have now specified that SD all are provided in Table 1 as arithmetic SD values.
- Histograms showing the distribution of ECd, ENAG, and Eβ2M are provided in SM to show large variance in these parameters.
- Table 2: For Log[(ENAG/Ccr)×103], U/L filtrate, it should be clarified whether the base was 10 or 2 or other. Also, in the results, it should be stated how the results are interpreted, e.g. how many times the NAG increases with increasing age by one year.
RESPONSE:
- Old Table 2 has now been changed to Table 4. The base of logarithmic transformation in Table 4 has been indicated.
- Estimated size of age effects on ENAG have been explained by standardized β values and unstandardized β values quoted below (lines 234-240).
Age showed an inverse association with ENAG in females (β = −0.170), and the high Cd burden group (β = −0.124), not in males or the low-Cd burden group. Per one year increase in age, log10 [(ENAG/Ccr] ×103] dropped 0.007 U/L filtrate in females and 0.004 U/L filtrate in the high-Cd burden groups. These were based on the unstandardized β values of log10 (ENAG/Ccr) ×103 versus age in females and the high-Cd-burden groups to be – 0.007 and – 0.004, respectively. These data can be interpreted to suggest a decrease in number of tubular cells or net loss of tubular cells per nephron as age increased, especially among women and those with high-Cd burden after adjustment for potential confounders.
- Line 186: Table 3→Table 2?
RESPONSE: A correction has been undertaken.
- Lines 306-308: It is possible that there is insufficient power to detect non-significant results in this study, and the interpretation that there is little or no association is an oversimplification.
RESPONSE:
- An analysis of an additional marker of tubular effect, namely Eβ2M, has been undertaken to obtain supporting evidence for the referred statement (lines 178-220).
- Thorough and meaningful explanation of all findings have been provided (lines 396-419), quoted above.
- Lines 313-315: This statement is not appropriate as the variation in NAG in this study cannot be solely attributed to Cd exposure.
RESPONSE: The referred statement has been deleted.
- Lines 323-324: ‘ECd/Ecr of 0.01-0.02 µg/g creatinine’is too low. It indicates normally undetectable levels, which may indicate a conversion error.
RESPONSE: The referred figures have been deleted.
- Lines 340-341: This is not a valid statement, as no arguments have been presented in this manuscript to deny the present evaluation that urinary Cd is good indicator of Cd body burden.
RESPONSE:
- The referred statements have been deleted.